# Dietary risk factors for hip fracture in adults: An umbrella review of meta-analyses of prospective cohort studies

**James Webster** [1] *, **Catherine E. Rycroft** [1], **Darren C. Greenwood** [2]⦾, **Janet E. Cade** [1]⦾

**1** School of Food Science and Nutrition, University of Leeds, Leeds, United Kingdom, **2** School of Medicine, University of Leeds, Leeds, United Kingdom

⦾ These authors contributed equally to this work.
* sp16jw@leeds.ac.uk

## Abstract

### Aim

To summarise the totality of evidence regarding dietary risk factors for hip fracture in adults, evaluating the quality of evidence, to provide recommendations for practice and further research.

### Design

Systematic review of meta-analyses of prospective cohort studies.

### Eligibility criteria

Systematic reviews with meta-analyses reporting summary risk estimates for associations between hip fracture incidence and dietary exposures including oral intake of a food, food group, beverage, or nutrient, or adherence to dietary patterns.

### Information sources

Medline, Embase, Web of Science, and the Cochrane Library from inception until November 2020.

### Data synthesis

The methodological quality of systematic reviews and meta-analyses was assessed using AMSTAR-2, and the quality of evidence for each association was assessed using GRADE. Results were synthesised descriptively.

### Results

Sixteen systematic reviews were identified, covering thirty-four exposures, including dietary patterns (n = 2 meta-analyses), foods, food groups, or beverages (n = 16), macronutrients (n = 3), and micronutrients (n = 13). Identified meta-analyses included 6,282 to 3,730,424 participants with between 322 and 26,168 hip fractures. The methodological quality

**Funding:** The authors received no specific funding for this work.

**Competing interests:** The authors have declared that no competing interests exist.

(AMSTAR-2) of all systematic reviews was low or critically low. The quality of evidence (GRADE) was low for an inverse association between hip fracture incidence and intake of fruits and vegetables combined (adjusted summary relative risk for higher vs lower intakes: 0.92 [95% confidence interval: 0.87 to 0.98]), and very low for the remaining thirty-three exposures.

## Conclusion

Dietary factors may play a role in the primary prevention of hip fracture, but the methodological quality of systematic reviews and meta-analyses was below international standards, and there was a lack of high-quality evidence. More long-term cohort studies reporting absolute risks and robust, well-conducted meta-analyses with dose-response information are needed before policy guidelines can be formed.

## Systematic review registration

PROSPERO CRD42020226190.

## Introduction

Fragility fracture is a global health issue that predominantly occurs in elderly populations [1]. Hip fracture in particular affects 18% of women and 6% of men globally; because the global population continues to age and grow, cases are projected to rise from 1.26 million in 1990 to 4.5 million by 2050 [2]. Hip fracture patients are at increased risk for other health problems, such as a decreased quality of life due to impaired mobility, and increased morbidity and mortality [1, 3, 4]. The economic burden is also high because of direct costs due to long hospitalisation and rehabilitation periods, and additional indirect costs associated with comorbidities [2]. Preventing hip fracture is therefore imperative to global public health clinically and economically.

Both genetic and environmental components contribute to the risk of hip fracture, including the potential for risk reduction through diet modification [3]. Associations between an array of dietary factors, including dietary patterns, foods, food groups, beverages, macronutrients, and micronutrients and hip fracture incidence have been the subject of previously published systematic reviews and meta-analyses. However, their methodological quality and the quality of evidence for most dietary factor-hip fracture associations are uncertain.

Umbrella reviews are one way of synthesising and critically appraising the quality of evidence from systematic reviews and meta-analyses to provide a comprehensive overview of a given topic with recommendations for practice or further research [5]. One umbrella review published in 2007 has synthesised the generic risk factors for hip fracture, including dietary factors [3]. Calcium combined with vitamin D supplementation and increasing dietary protein and tea intake decreased the risk of hip fracture, whilst alcohol, vitamin A, and caffeine intake increased the risk. However, their evaluation of the quality of evidence for these associations was based only on the consistency of results, and did not account for potential biases or imprecision. A recent scoping review also synthesised the evidence for non-pharmacological interventions in preventing hip fracture, but evidence for diet was restricted to nutritional supplementation in older adults, with little to no evidence on dietary patterns or dietary intake of foods or nutrients and hip fracture risk [4]. Therefore, we aimed to summarise the totality

of evidence regarding dietary risk factors for hip fracture in adults, evaluating the quality of evidence, to provide recommendations for practice or further research.

## Methods

The review protocol was registered in the International Prospective Register of Systematic Reviews (CRD42020226190). We followed the latest Preferred Reporting Items for Systematic Reviews and Meta-Analyses (PRISMA) guidelines (S7 and S8 Tables), and adhered to the guidance suggested by Fusar-Poli and Radua (2018) for the conduct of umbrella reviews [6, 7].

### Eligibility criteria

Articles were eligible if they were peer-reviewed and written in the English language. Eligible study designs were systematic reviews with meta-analysis of prospective cohort studies. Meta-analyses in any adult population (> 18 years) of any ethnicity, sex, or country were eligible. We included meta-analyses that pooled relative risks (RR), odds ratios (OR), or hazard ratios (HR) from studies assessing the relationship between a given dietary exposure and hip fracture incidence, where the dietary exposure was oral intake of a food, food group, beverage, or nutrient, or adherence to dietary patterns, such as the Mediterranean diet (MD).

Articles were excluded if they were duplicates, non-review articles, umbrella reviews, cohort pooling projects, or lacked a meta-analysis component (where no summary effect estimate was reported). Abstracts and conference proceedings were excluded. When a meta-analysis had one or more updates, the latest version was included and earlier versions were excluded, since the updated version usually contains more primary studies. Meta-analyses including case-control or cross-sectional studies were excluded unless a summary effect estimate was obtainable from cohort studies only. Cross-sectional and case-control studies are especially prone to selection and recall bias, thus were excluded to increase the quality of observational evidence. Meta-analyses where the population were non-adults, animal subjects, or were restricted to a specific patient population, such as diabetic or osteoarthritic patients, were excluded to enable wider generalisation of findings. We also excluded articles that assessed total fracture as the outcome without reporting a summary effect estimate for hip fracture. Meta-analyses of non-dietary exposures including supplements or exposures measured using biomarkers rather than dietary intake were excluded, such as those associating serum vitamin D levels with hip fracture incidence. Meta-analyses assessing hip fracture treatment or aftercare rather than prevention or risk were excluded to keep in line with our objectives.

### Information sources and search strategy

A systematic literature search of Medline, Embase, Web of Science, and the Cochrane Library of Systematic Reviews from inception until November 2020 was conducted. The search strategy comprised dietary exposure terms (foods, food groups, beverages, nutrients, and dietary patterns) combined with outcome (hip fracture) and review terms. The full search strategy is detailed in S1 Table. Returned umbrella reviews and reference lists of eligible full texts were scanned for potentially relevant articles. We did not search the grey literature or trial registries for additional articles.

### Screening and study selection

Two reviewers (JW and CR) independently screened the title and abstract of returned articles, and subsequently reviewed potentially relevant full-text articles for eligibility according to the pre-specified eligibility criteria. Discrepancies were resolved by consensus. Agreement

between authors regarding eligible articles was assessed using Cohen's kappa statistic, and was interpreted using the Altman scale [8]. In cases where multiple meta-analyses addressed the same association, we selected and included only the highest quality meta-analysis for each association identified based on our quality assessment to minimise risk of bias and to prevent double-counting primary studies. If multiple meta-analyses were judged to be of equal quality (had the same overall methodological quality and number of critical and non-critical flaws), the one with dose-response information was selected. If multiple meta-analyses still remained for one association, the meta-analysis with the greater number of total participants was selected for inclusion. The remaining eligible meta-analyses were excluded. The screening process was managed using EndNote (X9).

## Data extraction

Outcome data was independently extracted from eligible articles by two reviewers (JW and CR), with 83% agreement, and disagreements resolved by consensus. Study characteristics were extracted by one reviewer (JW). The characteristics, design, and exposure and outcome details of eligible systematic reviews were extracted. Review characteristics included: first author name, date of publication, number of primary studies included and their design, population characteristics, number of total subjects, number of subjects in each exposure category if reported, and the range in follow-up durations. Information extracted concerning the review and meta-analytic design included: number of databases searched with date ranges, objectives, methods used for meta-analysis (fixed or random-effects models), and sources of funding. Exposure details extracted from each review included: dietary exposures studied, dietary assessment methods used across primary studies per exposure (such as validated or non-validated food frequency questionnaires; FFQs), and the type of comparisons made with cut-off points used for categorical comparisons (i.e., high vs low) and increments used for linear dose-response comparisons per exposure. If categorical and dose-response comparisons were reported, only the latter was extracted. Outcome details extracted were: hip fracture assessment methods used across primary studies (such as self-reported questionnaires or review of medical records), total number of incident hip fracture cases at the latest point of follow-up, maximally adjusted summary effect estimates (OR, HR, or RR) with 95% confidence intervals (95% CI), estimates of heterogeneity, risk of small study effects, and confounders included in multivariable models of primary studies. Full risk of bias assessments were also extracted from each review at the domain level, and review authors were contacted for this data if it was not reported.

## Quality assessment

Two reviewers (JW and CR) independently assessed the methodological quality of eligible systematic reviews with meta-analyses using the validated AMSTAR-2 tool (a revised measurement tool to assess the methodological quality of systematic reviews) [9]. Discrepancies were resolved by consensus. Overall quality scores per review were determined by the number of critical and non-critical weaknesses. Of the domains covered by AMSTAR-2, those considered critical were: establishment of an *a priori* protocol (item 2), adequacy of the literature search (item 4), justification for excluding studies (item 7), adequate risk of bias assessment (item 9), appropriate meta-analytic methods (item 11), consideration of risk of bias when interpreting results (item 13), and risk of publication bias (item 15) [9]. Item 9 was considered a critical flaw if risk of bias assessments were not presented in full. Item 11 was considered a critical flaw if meta-analyses lacked dose-response information despite being appropriate, or combined effect sizes inappropriately. The methodological quality of reviews was considered high (< 1

non-critical weakness), moderate ($>$ 1 non-critical weakness), low (1 critical weakness with or without non-critical weaknesses), or critically low ($>$ 1 critical weakness with or without non-critical weaknesses) [9].

The quality of evidence for each dietary exposure-hip fracture association was then evaluated by applying the Cochrane GRADE tool (Grading of Recommendations for Assessment, Developing and Evaluation) based on evidence from the included meta-analyses (one per association) using the GRADEpro software and adhering to the GRADE handbook [10, 11]. Evidence was classified as high, moderate, low, or very low quality. If review authors did not present risk of bias assessments of primary studies in full, a serious risk was assumed. If no risk of bias assessment was reported, a very serious risk was assumed.

### Data synthesis

The findings of included meta-analyses (one per association) were tabulated and synthesised descriptively. We present adjusted summary effect estimates (RR, OR, or HR) with 95% CIs as reported in each meta-analysis, with $I^2$ values for heterogeneity, and Egger's p values for risk of publication bias.

## Results

### Study selection

The systematic search retrieved 841 publications, and two additional studies were found manually (Fig 1) [12, 13]. Of 601 unique articles, 138 full-texts were screened, and twenty-eight systematic reviews met the inclusion criteria, including sixty-two summary effect sizes. Agreement between authors regarding eligible articles was moderate (Cohen's k = 0.46). A list of excluded articles with reasons for their exclusion is documented in S2 Table. After selecting one meta-analysis per identified dietary exposure, sixteen systematic reviews with thirty-four summary effect estimates on dietary patterns (n = 2), foods, food groups, or beverages (n = 16), macronutrients (n = 3), and micronutrients (n = 13) remained and were included in the descriptive analysis regarding risk of hip fracture.

### Characteristics of eligible studies

Characteristics and findings of the twenty-eight eligible reviews are shown in S3 Table. Thirty-four unique exposures were identified, including: adherence to the alternative healthy eating index (AHEI) [14], adherence to the MD [15], dairy [16–18], milk [16–20], yogurt [16, 18, 20, 21], cheese [16, 18, 20, 21], fruits [22], vegetables [22], fruits and vegetables combined [22, 23], tea [24], coffee [13, 24, 25], alcohol (any, light, moderate, and heavy) [26], wine [26], beer [26], liquor (spirits) [26], total protein [27–29], animal protein [27, 29], vegetable protein [27, 29], dietary calcium [30–33], vitamin C [34, 35], vitamin A [36, 37], carotenoids [38], retinol [36, 37], a-carotene [38], b-carotene [36–38], b-cryptoxanthin [38], lycopene [38], lutein/zeaxanthin [38], alpha lipoic acid (ALA) [39], eicosapentaenoic acid with docosahexaenoic acid (EPA with DHA) [39], and antioxidant vitamin intake [40]. Multiple meta-analyses were retrieved for fourteen exposures, with only the highest quality meta-analysis retained per exposure. Only one meta-analysis was retrieved for: AHEI adherence [14], MD adherence [15], fruits [22], vegetables [22], tea [24], all alcohol associations [26], carotenoids [38], a-carotene [38], b-cryptoxanthin [38], lycopene [38], lutein/zeaxanthin [38], ALA [39], EPA with DHA [39], and antioxidant vitamin intake [40]. Most meta-analyses that investigated the same exposure included the same primary studies, with differences attributable to varying years of publication and eligibility criterion amongst meta-analyses. Nineteen systematic reviews reported their

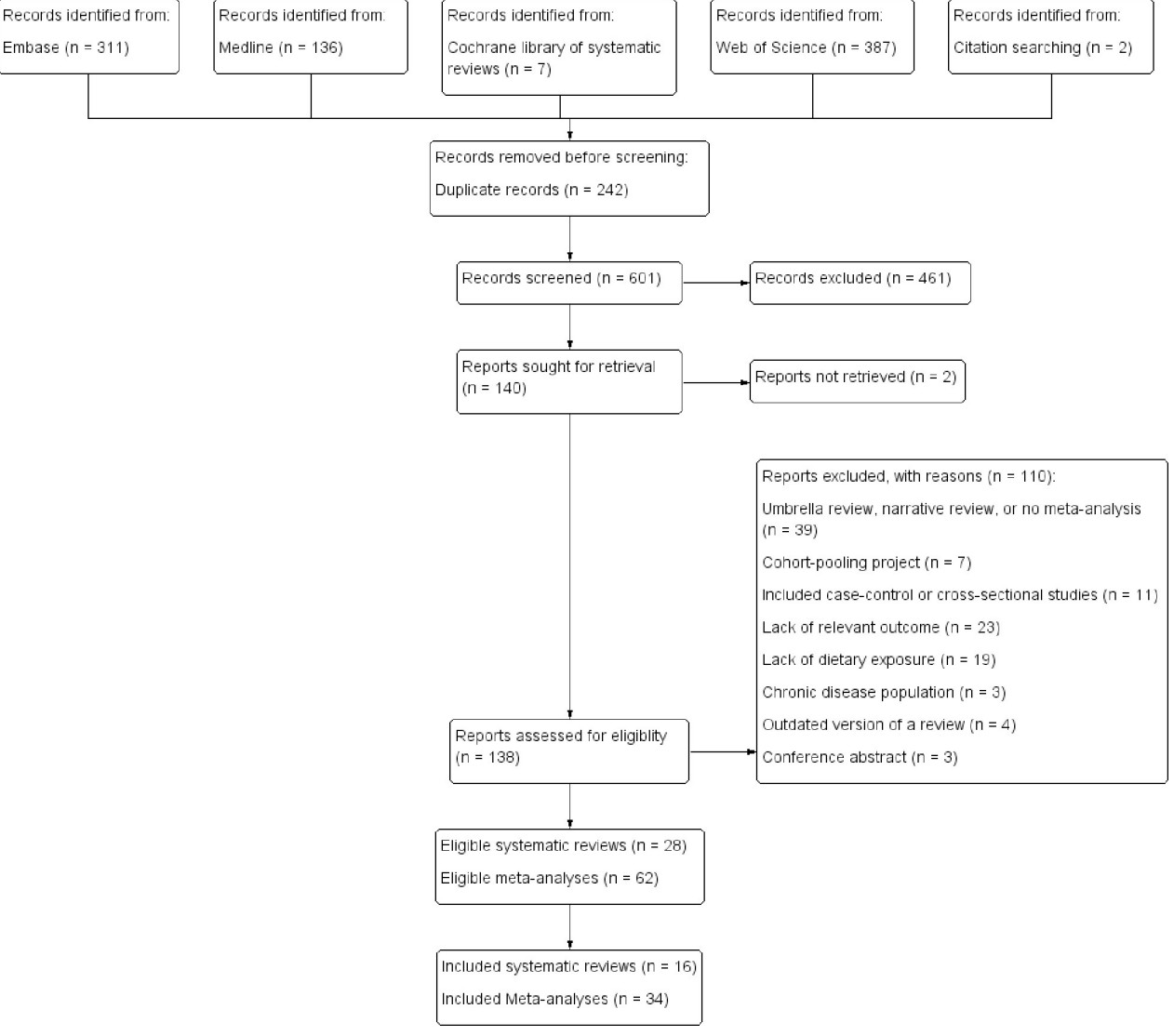

**Fig 1. PRISMA flow diagram of the study selection process.**

sources of funding, three declared no funding [21, 25], and six provided no information on funding [17, 22, 31, 33, 34, 39].

## Characteristics of included studies

When restricting analyses to one meta-analysis per association, the number of primary studies ranged from 2 to 18, with follow-up durations of 3 to 32 years. The total sample size in each meta-analysis ranged from 6,282 to 3,730,424 participants with 322 to 26,168 cases. Twenty-nine meta-analyses compared categories of consumption (e.g., high vs low), and five provided linear dose-response information [13, 15, 18, 20, 30]. The most common exposure measurement methods were validated or non-validated FFQs. Other techniques included 24-hour recalls and 7-day food records. Four meta-analyses provided no exposure measurement method information [27, 40]. Hip fracture was ascertained mostly by self-report (questionnaires or interviews) and review of medical records. Other techniques included radiologic or

x-ray exams, or linkage to hospital registers (such as the National Danish Patient Registry). Seven meta-analyses did not report hip fracture ascertainment techniques used in primary studies [23, 27, 30, 40]. Most meta-analyses included middle-aged to older adults of any sex or ethnicity. One meta-analysis regarding dairy intake was conducted in healthy non-Hispanic whites [18]. Three meta-analyses regarding ALA, EPA with DHA, and antioxidant vitamin intake excluded participants with a history of fracture [39, 40]. whilst the remaining studies did not report prior fracture as an exclusion criteria. Publication dates ranged from 2007 to 2020.

Each meta-analysis included cohort studies that adjusted for a variety of confounders (S3 Table). Summary effect estimates were adjusted for confounders in either all primary cohort studies in a meta-analysis (fully adjusted), or in some of the included primary cohort studies (partly adjusted). Confounders that were fully adjusted for included: age (n = 26 meta-analyses), smoking (n = 14), Body mass Index (BMI; n = 13), total energy intake (n = 11), physical activity (n = 10), calcium intake (n = 5), calcium and vitamin D supplementation (n = 3), alcohol intake (n = 2), hormone replacement therapy (n = 1), and height (n = 1). All meta-analyses except one included studies adjusted for or stratified by sex, or included single-sex studies [23]. Four meta-analyses presented summary effect estimates that were not fully adjusted for any potential confounders, corresponding to the following exposures: adherence to the AHEI [14], milk [20], protein [27], and dietary calcium [30]. Summary effect estimates were often partly adjusted for weight, protein intake, caffeine intake, history of fracture or fall, and chronic disease.

## Methodological quality

The overall and item-specific AMSTAR-2 ratings for each eligible systematic review and their meta-analyses are shown in S5 Table. Of the sixteen included systematic reviews, the methodological quality was low in one [20], and critically low in the remaining fifteen reviews. Three summary effect estimates were extracted from the low-quality review regarding milk, yogurt, and cheese consumption based on cohort studies [20]. Effect estimates for the remaining dietary exposures were reported from critically low-quality reviews. Common critical weaknesses in reviews were that methods were not established *a priori*; excluded studies were not listed with justification for their exclusion; risk of bias assessments were mostly inadequate; and meta-analytic methods used were often inappropriate, such as when meta-analyses compared 'high vs low' dietary exposure categories using varying thresholds for these categories from each primary cohort study. All risk of bias assessments used the Newcastle-Ottawa scale, which does not assess the potential for bias due to selective reporting or changes in dietary exposure classifications throughout follow-up [41]. Nine included reviews did not report domain-specific risk of bias assessments [14, 15, 22, 34, 37–40], and four reviews did not report any risk of bias assessment [13, 24, 26, 30].

## Dietary exposures and hip fracture: Associations and quality of evidence

Adjusted summary effect estimates and the quality of evidence for each exposure regarding risk of hip fracture are summarised in Tables 1–4. Of thirty-four potential diet-hip fracture associations, the quality of evidence for each was graded as low (n = 1) [23] and very low (n = 33), respectively. No association was considered as moderate or high-quality evidence. The quality of evidence for all associations except combined fruits and vegetables intake and hip fracture incidence was downgraded from low to very low due to a serious or very serious risk of bias amongst primary studies [23]. For sixteen associations, a serious risk of bias was assumed because their meta-analyses did not report risk of bias assessments at the domain

**Table 1. Summary characteristics and findings of meta-analyses of cohort studies assessing associations between dietary patterns and hip fracture risk.**

| Exposure | Author (year) | n studies | Follow-up range (years) | Comparison | Summary effect estimate (95% CI) | I² (%) | Egger's p-value | AMSTAR | GRADE |
|---|---|---|---|---|---|---|---|---|---|
| Alternative healthy eating index | Panahande et al. (2018) | 4 | 10–32 | High vs low | RR: 0.83 (0.71, 0.97) | N/A | 0.19 | Critically low | ⊕○○○ |
| Mediterranean diet | Malmir et al. (2018a) | 4 | 8–16 | Per 1 score increase in adherence | RR: 0.95 (0.92, 0.98) | 68* | 0.78 | Critically low | ⊕○○○ |

N/A = not applicable or available

* = significant heterogeneity; RR = relative risk; 95% CI = 95% confidence interval; ⊕○○○ = very low quality of evidence.

**Table 2. Summary characteristics and findings of meta-analyses of cohort studies assessing associations between dietary intake of foods, food groups, and beverages and hip fracture risk.**

| Exposure | Author (year) | n studies | Follow-up range (years) | Comparison | Summary effect estimate (95% CI) | I² (%) | Egger's p-value | AMSTAR | GRADE |
|---|---|---|---|---|---|---|---|---|---|
| Total dairy | Matia-Martin et al. (2019) | 4 | 8–22 | Per 'increment' increase | RR: 0.98 (0.95, 1.01) | 86* | 0.98 | Critically low | ⊕○○○ |
| Milk | Hidayat et al. (2020) | 7 | 6–21 | Per 1 glass/day increase | RR: 0.97 (0.92, 1.03) | 60* | 0.21 | Low | ⊕○○○ |
| Yogurt | Hidayat et al. (2020) | 4 | 12–21 | High vs low | RR: 0.78 (0.68, 0.90) | 14 | > 0.45 | Low | ⊕○○○ |
| Cheese | Hidayat et al. (2020) | 4 | 6–21 | High vs low | RR: 0.85 (0.66, 1.08) | 77* | > 0.45 | Low | ⊕○○○ |
| Fruits | Luo et al. (2016) | 5 | 8–14 | High vs low | HR: 0.91 (0.77, 1.07) | 73* | N/A | Critically low | ⊕○○○ |
| Vegetables | Luo et al. (2016) | 5 | 8–14 | High vs low | HR: 0.81 (0.68, 0.96) | 71* | N/A | Critically low | ⊕○○○ |
| Fruit and vegetables | Brondani et al. (2019) | 5 | 7–20 | High vs low | RR: 0.92 (0.87, 0.98) | 56 | 0.15 | Critically low | ⊕⊕○○ |
| Tea | Sheng et al. (2013) | 3 | 6–12 | High vs low | RR: 1.03 (0.54, 1.52) | 42 | 0.06 | Critically low | ⊕○○○ |
| Coffee | Li and Xu (2013) | 4 | 6–30 | Per cup increase/day | OR: 1.00 (0.96, 1.03) | N/A | 0.89 | Critically low | ⊕○○○ |
| Alcohol | Zhang et al. (2015) | 18 | 3–30 | Any vs none | RR: 1.03 (0.91, 1.15) | 72* | > 0.10 | Critically low | ⊕○○○ |
|  | Zhang et al. (2015) | 7 | 3–30 | Light vs none | RR: 0.88 (0.83, 0.92) | 20 | > 0.10 | Critically low | ⊕○○○ |
|  | Zhang et al. (2015) | 7 | 3–30 | Moderate vs none | RR: 1.00 (0.85, 1.14) | 56* | > 0.10 | Critically low | ⊕○○○ |
|  | Zhang et al. (2015) | 3 | 3–30 | Heavy vs none | RR: 1.71 (1.41, 2.01) | 0 | > 0.10 | Critically low | ⊕○○○ |
| Wine | Zhang et al. (2015) | 4 | 3–14 | Any vs no alcohol | RR: 0.81 (0.71, 0.92) | 0 | > 0.10 | Critically low | ⊕○○○ |
| Beer | Zhang et al. (2015) | 4 | 3–14 | Any vs no alcohol | RR: 1.13 (0.69, 1.56) | 79* | > 0.10 | Critically low | ⊕○○○ |
| Liquor (spirits) | Zhang et al. (2015) | 4 | 3–14 | Any vs no alcohol | RR: 0.94 (0.75, 1.12) | 33 | > 0.10 | Critically low | ⊕○○○ |

N/A = not applicable or available

* = significant heterogeneity; RR = relative risk; OR = odds ratio; HR = hazard ratio; 95% CI = 95% confidence interval; ⊕○○○ = very low quality of evidence; ⊕⊕○○ = low quality of evidence. For milk and coffee consumption, I² and Egger's p-value were unobtainable from dose-response meta-analyses, thus values from high vs low comparisons are presented as an estimate where available.

**Table 3. Summary characteristics and findings of meta-analyses of cohort studies assessing associations between dietary intake of macronutrients and hip fracture risk.**

| Exposure | Author (year) | n studies | Follow-up range (years) | Comparison | Summary effect estimate (95% CI) | I² (%) | Egger's p-value | AMSTAR | GRADE |
|---|---|---|---|---|---|---|---|---|---|
| Dietary protein | Wu et al. (2015) | 3 | N/A | High vs low | RR: 0.89 (0.82, 0.97) | 0 | 0.05 | Critically low | ⊕○○○ |
| Animal protein | Wu et al. (2015) | 4 | N/A | High vs low | RR: 1.04 (0.70, 1.54) | 52 | 0.90 | Critically low | ⊕○○○ |
| Vegetable protein | Wu et al. (2015) | 3 | N/A | High vs low | RR: 1.00 (0.53, 1.91) | 57 | 0.91 | Critically low | ⊕○○○ |

N/A = not applicable or available; RR = relative risk; 95% CI = 95% confidence interval; ⊕○○○ = very low quality of evidence.

level (i.e., potential biases from specific sources). Twenty-two associations were further downgraded due to inconsistency (n = 8), imprecision (n = 8), or both (n = 6). No association was upgraded in quality for any reason. Full quality of evidence assessments using GRADE are shown in S6 Table.

**Dietary patterns.** Table 1 shows the findings and the quality of evidence for associations between adherence to dietary patterns and the incidence of hip fracture. An inverse linear

**Table 4. Summary characteristics and findings of meta-analyses of cohort studies assessing associations between dietary intake of micronutrients and hip fracture risk.**

| Exposure | Author (year) | n studies | Follow-up range (years) | Comparison | Summary effect estimate (95% CI) | I² (%) | Egger's p-value | AMSTAR | GRADE |
|---|---|---|---|---|---|---|---|---|---|
| Dietary calcium intake | Bischoff-Ferrari et al. (2007) | 4 | 3–18 | Per 300 mg increase/day | RR: 1.01 (0.96, 1.06) | N/A | N/A | Critically low | ⊕○○○ |
| Dietary vitamin C | Malmir et al. (2018b) | 3 | 13–15 | High vs low | RR: 0.92 (0.59, 1.44) | 55 | 0.83 | Critically low | ⊕○○○ |
| Dietary vitamin A | Zhang et al. (2017) | 3 | 12–18 | High vs low | RR: 1.29 (1.06, 1.57) | 0 | 0.85 | Critically low | ⊕○○○ |
| Dietary carotenoids | Xu et al. (2017) | 2 | 10–17 | High vs low | OR: 0.72 (0.51, 1.01) | 59 | 0.16 | Critically low | ⊕○○○ |
| Dietary retinol | Zhang et al. (2017) | 4 | 12–18 | High vs low | RR: 1.40 (1.02, 1.91) | 65* | 0.17 | Critically low | ⊕○○○ |
| Dietary a-carotene | Xu et al. (2017) | 2 | 10–17 | High vs low | OR: 0.77 (0.55, 1.08) | 64* | 0.36 | Critically low | ⊕○○○ |
| Dietary b-carotene | Zhang et al. (2017) | 2 | 17–18 | High vs low | RR: 0.91 (0.64, 1.31) | 82* | 0.80 | Critically low | ⊕○○○ |
| Dietary b-cryptoxanthin | Xu et al. (2017) | 2 | 10–17 | High vs low | OR: 1.11 (0.97, 1.28) | 0 | 0.49 | Critically low | ⊕○○○ |
| Dietary lycopene | Xu et al. (2017) | 2 | 10–17 | High vs low | OR: 0.84 (0.69, 1.01) | 8 | 0.14 | Critically low | ⊕○○○ |
| Dietary lutein/ zeaxanthin | Xu et al. (2017) | 2 | 10–17 | High vs low | OR: 0.94 (0.79, 1.11) | 8 | 0.60 | Critically low | ⊕○○○ |
| Dietary ALA | Sadheghi et al. (2019) | 3 | 8–24 | High vs low | RR: 1.01 (0.90, 1.13) | 71* | N/A | Critically low | ⊕○○○ |
| Dietary EPA + DHA | Sadheghi et al. (2019) | 3 | 8–24 | High vs low | RR: 0.91 (0.81, 1.03) | 0 | N/A | Critically low | ⊕○○○ |
| Antioxidant vitamins | Zhou et al. (2020) | 9 | 4–19 | High vs low | RR: 0.87 (0.69, 1.08) | 89* | 0.45 | Critically low | ⊕○○○ |

N/A = not applicable or available

* = significant heterogeneity; RR = relative risk; OR = odds ratio; 95% CI = 95% confidence interval; ⊕○○○ = very low quality of evidence; For antioxidant vitamins intake, Egger's p-value was obtained for total fracture, since this value was not presented for just hip fracture and both outcomes included mostly the same primary studies.

association was observed between MD adherence and hip fracture incidence from very low-quality evidence (for an increment of one unit in MD score, adjusted summary HR: 0.95 (95% CI: 0.92, 0.98)) [15]. AHEI adherence was also inversely associated with hip fracture incidence in a higher vs lower comparison based on very low-quality evidence (RR: 0.83 (0.71, 0.97)) [14]. No other associations were identified regarding dietary patterns and hip fracture.

**Foods, food groups, and beverages.** Table 2 shows the findings and the quality of evidence for associations between intake of foods, food groups, and beverages and the incidence of hip fracture. Low-quality evidence was found for an inverse association between intake of fruits and vegetables combined and hip fracture when comparing higher vs lower intakes (HR: 0.92 (0.87, 0.98)) [23]. An inverse association was also observed for higher vs lower vegetable and yogurt intakes and hip fracture incidence with very low-quality evidence, respectively (vegetables HR: 0.81 (0.68, 0.96); yogurt RR: 0.78 (0.68, 0.90)) [20, 22]. For alcohol intake, multiple comparisons were made with very low-quality evidence: compared to abstainers, light alcohol intake (0.01–12.5 g/day) and wine intake were inversely associated with hip fracture (RR: 0.88 (0.83, 0.92); RR: 0.81 (0.71, 0.92)) [26]. Conversely, heavy (> 50 g/day) vs no alcohol intake was positively associated with hip fracture (RR: 1.71 (1.41, 2.01)) [26]. No significant association was observed for any or moderate alcohol intake (12.6–49.9 g/day), beer, and liquor vs no alcohol and hip fracture [26]. Very low quality evidence also showed no clear association between hip fracture and total dairy [18], milk [20], fruits [22], cheese [20], tea [24], and coffee intakes [13].

**Macronutrients.** Table 3 shows the findings and the quality of evidence for associations between intake of macronutrients and the incidence of hip fracture. Very low-quality evidence showed an inverse association between total dietary protein intake and hip fracture incidence (higher vs lower RR: 0.89 (0.82, 0.97)) [27]. No clear association was observed between animal or vegetable protein intake and hip fracture in a higher vs lower comparison of very low-quality evidence, respectively [27].

**Micronutrients.** Table 4 shows the findings and the quality of evidence for associations between the intake of micronutrients and the incidence of hip fracture. The quality of evidence was very low for all micronutrient exposures. Near-significant inverse associations were observed between dietary intake of carotenoids or lycopene and hip fracture (higher vs lower intakes; carotenoids OR: 0.72 (0.51, 1.01); lycopene OR: 0.84 (0.69, 1.01)) [38]. Significant positive associations were observed between dietary vitamin A and retinol intake and hip fracture in meta-analyses of higher vs lower comparisons, respectively (vitamin A RR: 1.29 (1.06, 1.57); retinol RR: 1.40 (1.02, 1.91)) [37]. In dose-response meta-analyses (per 300 mg increase per day), no clear association was observed between hip fracture and dietary calcium intake [30]. In meta-analyses comparing higher vs lower intakes, no clear association was observed between hip fracture and dietary intake of vitamin C [34], a-carotene [38], b-carotene [37], b-cryptoxanthin [38], lutein/zeaxanthin [38], ALA [39], EPA with DHA [39], or antioxidant vitamins [40].

**Heterogeneity.** Estimates of the proportion of heterogeneity attributable to between-study variation using the $I^2$ statistic were available in all but three meta-analyses [13, 14, 30]. Nineteen meta-analyses had an $I^2$ value above 50%, and heterogeneity was significant for the following 14 exposures: MD adherence [15], total dairy [18], milk [20], cheese [20], fruits [22], vegetables [22], alcohol (any, moderate, beer) [26], retinol [37], a-carotene [38], b-carotene [37], ALA [39], and antioxidant vitamins [40]. $I^2$ values were above 75% for total dairy [18], cheese [20], beer [26], b-carotene [37], and antioxidant vitamin intake [40]. $I^2$ values were not available for AHEI adherence [14], coffee intake [13], and dietary calcium intake [30].

**Small study effects.** All meta-analyses assessed the risk of small study effects (such as publication bias) except those regarding fruits [22], vegetables [22], animal protein [27], and

vegetable protein intake [27]. Methods used included: Egger's test (n = 30 meta-analyses); Egger's and Begg's test (n = 21); both tests with visual inspection of funnel plots (n = 11); and Egger's test with visual inspection (n = 7). No meta-analysis showed evidence of publication bias.

## Discussion

### Principal findings

The effects of several dietary factors on the incidence of hip fracture have been quantified in previously published meta-analyses. This umbrella review aimed to summarise the totality of evidence regarding diet and hip fracture risk in adults, evaluating the quality of evidence for each association. We included thirty-four meta-analyses of cohort studies relating different dietary factors to hip fracture incidence. The key findings of this umbrella review are four-fold. Firstly, low-quality evidence showed that high intake of fruits and vegetables combined may decrease the risk of hip fracture compared to low intakes [23]. Secondly, there was no clear association between consumption of dairy, dairy products, or dietary calcium and hip fracture incidence based on very low-quality evidence [18, 20, 30]. Thirdly, there was a lack of evidence regarding dietary patterns and hip fracture incidence. Finally, the methodological quality of most systematic reviews and their meta-analyses and the quality of evidence for most diet-hip fracture associations were very low.

### Comparison with other studies

Few up-to-date, evidence-based guidelines exist for preventing hip fracture through diet [3, 42]. Our umbrella review addresses the dietary risk factors for hip fracture identified in a previous umbrella review (excluding nutritional supplements) [3], and provides information for 31 additional dietary exposures.

The National Osteoporosis Foundation (NOF) and a previous umbrella review suggest increasing consumption of fruits and vegetables or vegetables alone to prevent hip fracture [3, 42, 43]. Consistent with this, we found that higher intake of vegetables but not fruits was associated with a reduced risk of hip fracture, but with very low-quality evidence due to serious inconsistency and a serious risk of bias among primary studies [22]. We also found low-quality evidence that a higher intake of fruits and vegetables combined was associated with a reduced risk of hip fracture [23]. Further cohort studies with dose-response and absolute risk information would increase the certainty of evidence for this association. The potential for fruits or vegetables alone to reduce hip fracture incidence requires further research.

Contrary to existing recommendations [42], we found no clear association between hip fracture incidence and intake of dietary calcium, total dairy, and dairy products except yogurt, which was inversely associated with hip fracture. The quality of evidence for these associations was very low due to a serious risk of bias amongst primary cohort studies (stemming from inadequate follow-up data and unvalidated dietary assessment methods) and inconsistent results [18, 20]. Meta-analyses excluded from our review largely support these findings [16, 17, 19, 21, 31–33], though in higher vs lower comparisons, one meta-analysis found a near-significant protective effect of total dairy intake, and a reduced risk of hip fracture with higher cheese consumption [16]. This discrepancy may be due to differences in primary studies included and meta-analytic methods used between meta-analyses.

To our knowledge, no specific dietary pattern is recommended for the primary prevention of hip fracture. We found that higher adherence to the AHEI or the MD may decrease the risk of hip fracture, but the quality of evidence was very low for both associations due to a serious risk of bias amongst primary studies [14, 15]. Our findings were consistent with other meta-

analyses that included case-control studies [44, 45], such as a recent meta-analysis that showed that higher adherence to a diet high in fruits and vegetables, poultry, fish, and wholegrains (resembling the MD) was associated with a reduced risk of hip fracture, whilst higher adherence to a 'Western diet' (high red meat, processed meat, animal fat, eggs, and sweets) increased the risk [45]. Future large prospective cohort studies are needed to explore the effects of various dietary patterns on the risk of hip fracture before preventative recommendations can be made. A recent European Prospective Investigation into Cancer (EPIC) cohort study published after our last database search showed a greater risk of hip fracture among fish-eaters, vegetarians, and vegans compared to meat-eaters after adjustment for socio-economic factors, lifestyle confounders, and BMI in a UK population that consisted predominantly of middle-aged, white females [46]. Equivalent studies would help elucidate the effect of other dietary patterns on hip fracture incidence. This is of particular importance given that dietary patterns encompass several individual dietary risk factors that could act synergistically to impact the risk of hip fracture [47].

National Osteoporosis Foundation guidelines and previous umbrella reviews have suggested increasing consumption of tea and dietary protein, and limiting alcohol consumption to prevent hip fracture [3, 42, 48]. We also found a protective effect of dietary protein [27], but the effect of tea was unclear [24], and the association between alcohol intake and hip fracture was dependent on the type and amount consumed [26]. In line with previous evidence [24, 25], we also found no clear association between coffee consumption and hip fracture [13]. In any case, the methodological quality of included systematic reviews and meta-analyses and the quality of evidence for these associations were very low. Many meta-analyses assessing diet-hip fracture associations included a small number of studies (< 5 studies), reducing statistical power and the precision of confidence intervals. Many showed a serious risk of bias within included studies, presented an inadequate quality assessment (for which we assumed a serious risk of bias), or did not present a quality assessment of included studies (for which we assumed a very serious risk of bias). Several associations also showed a high degree of inconsistency. Future large, well-conducted cohort studies are required to remedy these issues.

Epidemiological evidence has considered the role of other dietary factors in relation to risk of hip fracture, such as dietary intake of carbohydrates, fats, B-vitamins, magnesium, zinc, and isoflavones [49–55]. These studies were not included in our review because they lacked a meta-analysis component, did not assess the effect of a dietary exposure on hip fracture risk, or met other aspects of our exclusion criteria. Nonetheless, their findings are relevant points for future research. For instance, one meta-analysis including case-control studies showed a strong, positive association between saturated fatty acid intake and hip fracture incidence (RR: 1.79 [1.05, 3.03]) [53]. Well-conducted trials, cohort studies, and meta-analyses are needed to better understand the role of dietary factors in preventing hip fracture, including those not explored in our review.

## Possible mechanisms

A possible explanation for the inverse association between intake of fruits and vegetables and hip fracture risk is the potential for fruits and vegetables to decrease bone loss through several potential pathways, which decreases fracture risk [23]. Potential pathways include shifting the acid-base balance to a more alkaline state to increase calcium reabsorption, and improving bone remodelling by reducing oxidative stress [23]. A third proposed mechanism is that fruits and vegetables may decrease chronic inflammation, which is associated with fracture incidence [23, 56].

Alternatively, those that consume more fruits and vegetables may have other healthy dietary and lifestyle habits, and may be less likely to have other health problems (such as diabetes or depression) that could increase the risk of hip fracture [43, 57]. Indeed, adherence to dietary patterns with higher intakes of fruits and vegetables (AHEI and the MD) reduced the risk of hip fracture in our review. The association between intake of fruits and vegetables and hip fracture incidence remained after adjustment for relevant confounders, but physical activity was not accounted for [23]. Since physical activity is positively associated with fruit and vegetable intake and inversely associated with hip fracture incidence, the apparent protective effect of fruits and vegetables against hip fracture could be exaggerated by residual confounding [58, 59].

Milk, yogurt, cheese, and therefore total dairy consumption could plausibly reduce the risk of hip fracture via their high content of nutrients associated with bone health acting synergistically, including protein, calcium, and vitamin D [18]. Milk, however, is a major source of D-galactose, which could contribute to bone loss through oxidative stress and chronic inflammation, thus may have no net effect on hip fracture incidence [20]. The potential benefit of yogurt but not cheese consumption may be explained by a lack of statistical power to detect an association for cheese and hip fracture, or the association between cheese consumption and hip fracture incidence may depend on the type of cheese [20]. Indeed, a large degree of unexplained heterogeneity was observed for cheese but not yogurt consumption [20]. Total dairy intake may therefore not be associated with hip fracture because of the large degree of heterogeneity between studies caused by combining effects of different dairy products into a single summary effect.

Associations between dairy, dairy products, dietary calcium intake, and hip fracture incidence may also depend on factors not considered in our review, further contributing to the heterogeneity observed for total dairy intake. The effect of dietary calcium intake could depend on sex, but the meta-analysis here was restricted to women [30]. Findings for total dairy were limited to non-Hispanic whites [18], but could vary by ethnicity. In their sub-group analyses, Hidayat et al. (2020) showed a protective effect for milk that was attenuated by adjustment for BMI and physical activity [20]. Moreover, milk had a protective effect in USA populations but not in Scandinavian populations, and was attributed to differences in vitamin D fortification policies for milk between the two regions [20]. Therefore, obtaining higher quality evidence regarding dairy and calcium intake and hip fracture incidence to enable strong preventative recommendations to be made is challenging. Future high-quality cohort studies should aim to clarify the relationship between calcium intake and individual dairy products and hip fracture incidence.

## Strengths and limitations

The main strength of this umbrella review was that it systematically synthesised the totality of evidence from all published meta-analyses of cohort studies considering the role of diet and dietary factors in preventing hip fracture. We used validated tools to assess the methodological quality of systematic reviews and meta-analyses and the overall quality of evidence for each association. Common methodological limitations of systematic reviews and meta-analyses and implications for further research were identified.

Given the broad scope of our umbrella review, we restricted our search to systematic reviews with meta-analyses. In doing so, cohort studies that were not included in a previously published meta-analysis may have been missed, such as the recently published EPIC-Oxford cohort study comparing dietary patterns and hip fracture risk [46]. These studies may point towards understudied dietary risk factors for hip fracture.

We did not re-meta-analyse primary study data from eligible meta-analyses for each association because the principal objective of an umbrella review is to summarise existing relevant research syntheses [60]. We relied on the information reported in included systematic reviews and meta-analyses when assessing the quality of evidence for each association using the GRADE tool; therefore, our review inherited their limitations. Many included systematic reviews did not present a risk of bias assessment at the domain level for each primary cohort study. Evidence for these associations were assumed to contain a serious or very serious risk of bias, downgrading their quality. As observational research begins at low quality in GRADE (due to being non-randomised), the quality of evidence for all but one association in our umbrella review was very low. Therefore, our review could have underestimated the quality of evidence for associations assessed, but given the high degree of inconsistency and imprecision of results across cohort studies for many associations assessed, this is unlikely. This could limit the ability of our umbrella review to provide evidence for the formulation of preventative recommendations against hip fracture, but highlights the need for rigorous risk of bias assessments in future systematic reviews to facilitate more robust assessments of the quality of evidence.

Our umbrella review identified other limitations common to meta-analyses of diet and hip fracture that reduced the quality of evidence for many associations. Knowledge of the absolute risk of hip fracture for a given dietary exposure is required to put relative effects into context, and for robust evidence-based recommendations to be made [61]. Since hip fracture is a rare event in cohort studies, large relative effects may not translate into large absolute effects. However, none of the included meta-analyses reported absolute effects for risk of hip fracture, and this could not be calculated due to insufficient data in meta-analyses (meta-analyses did not report baseline risks of hip fracture in their respective populations, and most did not report the number of participants in each exposure category).

Many meta-analyses lacked dose-response information and compared high vs low consumption of a dietary factor on hip fracture risk without defining thresholds for these categories. Since cohort studies often define exposure categories at different thresholds, pooling effect estimates from comparisons between different levels of an exposure masks the true comparison being made, and limits the utility of findings.

All included meta-analyses included observational studies. Excluding met-analyses of case-control or cross-sectional studies reduced the risk of recall and selection bias, and ensured that dietary assessments preceded hip fracture events. However, cohort studies remain prone to residual confounding bias. Most meta-analyses included cohort studies with effect estimates that were unadjusted for prior fracture–a risk factor for subsequent fracture–and did not explicitly exclude participants with a history of fracture [3]. Other key confounders that were often not adjusted for by cohort studies within meta-analyses include total energy intake, alcohol intake, and the presence of chronic disease. Therefore, summary effect estimates for most associations were not fully adjusted for key confounders, distorting true effect sizes.

## Conclusions and future directions

The effects of dietary factors on the risk of hip fracture in adults have been quantified in previously published meta-analyses. It is clear from these that foods, nutrients, and dietary patterns could play a role in the primary prevention of hip fracture. Fruits and vegetables intake may decrease the risk of hip fracture, but the methodological quality of almost all systematic reviews and meta-analyses was critically low, and the effects of other dietary exposures such as dairy and calcium intakes on hip fracture risk was uncertain. This umbrella review highlights the potential dietary risk factors for hip fracture that warrant further research, and points towards the need for review authors to improve the conduct and reporting of their syntheses.

Given the very low quality of evidence for most associations, we recommend more well-conducted, long-term cohort studies and subsequently robust meta-analyses so that stronger policy recommendations can be made to prevent hip fracture through diet. To increase the quality of evidence, authors of systematic reviews should establish their protocol *a priori*, justify the exclusion of each study, and provide a rigorous assessment of the risk of bias of included primary studies. Authors of meta-analyses should additionally present dose-response information and absolute risks if possible, explore sources of heterogeneity, and consider if meta-analysis is appropriate with the data available. We recommend a particular focus on the effect of dietary patterns to account for interactive effects of foods and nutrients on the risk of hip fracture.

## Supporting information

**S1 Table. Search strategy.**
(DOCX)

**S2 Table. Excluded articles with justifications for their exclusion.**
(DOCX)

**S3 Table. Characteristics of eligible meta-analyses assessing the association between risk of hip fracture and dietary exposures.**
(DOCX)

**S4 Table. Sources of funding for eligible systematic reviews.**
(DOCX)

**S5 Table. Methodological quality assessments of eligible systematic reviews using the AMSTAR-2 tool.**
(DOCX)

**S6 Table. Quality of evidence assessments per association from the highest quality meta-analysis per exposure using the GRADE tool.**
(DOCX)

**S7 Table. PRISMA 2020 checklist.**
(DOCX)

**S8 Table. PRISMA 2020 checklist for abstracts.**
(DOCX)

## Acknowledgments

We would like to thank Dr Dena Zaraatkar and Dr Russell de Souza for their advice regarding umbrella review methodology and use of the GRADE tool.

## Author Contributions

**Conceptualization:** Darren C. Greenwood, Janet E. Cade.

**Data curation:** James Webster.

**Formal analysis:** James Webster, Catherine E. Rycroft.

**Investigation:** James Webster.

**Methodology:** James Webster, Darren C. Greenwood, Janet E. Cade.

**Project administration:** James Webster.

**Supervision:** Darren C. Greenwood, Janet E. Cade.

**Visualization:** James Webster.

**Writing – original draft:** James Webster.

**Writing – review & editing:** James Webster, Catherine E. Rycroft, Darren C. Greenwood, Janet E. Cade.

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
