## [Decision Letter · Decision Letter 0]

24 Sep 2021

PONE-D-21-20733Dietary risk factors for hip fracture: an umbrella review of meta-analyses of prospective cohort studiesPLOS ONE

Dear Dr.  Webster,

Thank you for submitting your manuscript to PLOS ONE. After careful consideration, we feel that it has merit but does not fully meet PLOS ONE’s publication criteria as it currently stands. Therefore, we invite you to submit a revised version of the manuscript that addresses the points raised during the review process.

We look forward to receiving your revised manuscript.

Kind regards,

Yuan-Pin Hsu

Academic Editor

PLOS ONE

Journal Requirements:

2. Please note that in order to use the direct billing option the corresponding author must be affiliated with the chosen institute. Please either amend your manuscript to change the affiliation or corresponding author, or email us at plosone@plos.org with a request to remove this option.

Reviewers' comments:

Reviewer's Responses to Questions

**Comments to the Author**

1. Is the manuscript technically sound, and do the data support the conclusions?

Reviewer #1: Yes

Reviewer #2: Yes

2. Has the statistical analysis been performed appropriately and rigorously? 

Reviewer #1: Yes

Reviewer #2: Yes

3. Have the authors made all data underlying the findings in their manuscript fully available?

Reviewer #1: Yes

Reviewer #2: Yes

4. Is the manuscript presented in an intelligible fashion and written in standard English?

Reviewer #1: Yes

Reviewer #2: Yes

5. Review Comments to the Author

Reviewer #1: Title: `Dietary risk factors for hip fracture: an umbrella review of Meta-analyses of prospective cohort studies`

• Title:

In articulating a title of this particular Umbrella Review, the population of interest is not evident.

Is it a review of Meta-analysis (which is only 2 out of 34-exposures)?

Suggestion: Dietary risk factors for hip fracture in adults: An Umbrella Review of Prospective cohort studies.

• Objective:

`To summarize the totality of evidence regarding dietary factors and hip fracture incidence in adults, and to evaluate the methodological quality of systematic reviews and meta-analyses

Is it appropriate to use the Term `Incidence`?

`Methodological quality of systematic reviews` -This is the analysis procedure to be performed during the process and not the objective to be set

Suggestion: To summarize the `totality of evidence regarding dietary factors for hip fracture` in adults

• Methodology and synthesis:

``The quality of evidence for all associations except combined fruits and vegetables intake and hip fracture incidence was downgraded from low to very low due to a serious or very serious risk of bias amongst primary studies``

The purpose of UR quality assessment is to assess methodological quality, risk of bias, and reporting quality of reviews, and does not involve retrieving the quality of primary studies??

Any modifications to the JBI umbrella review method?

• Result;

`The quality of evidence (GRADE) was low for an inverse association between hip fracture incidence and intake of fruits and vegetables combined (adjusted relative risk for higher vs lower intakes: 0.92 [95% confidence interval: 0.87 to 0.98]), and very low for the remaining thirty-three exposures. Adjusted summary effect estimates and the quality of evidence for each exposure regarding risk of hip fracture are summarized.

Question:

Wasn`t it necessary to show cumulative risk factor (effect size)? By forest plot, Summary ROC (SROC) curves…or other ways of addressing heterogeneity of the Meta-analysis. Because, if the differences are not the result of chance, then we need to be cautious in interpreting the results of the meta-analysis.

Nineteen meta-analyses had an I2 value above 50%, and heterogeneity was significant for the 14 exposures. If studies are heterogeneous, is it certain or appropriate to synthesize the respective studies into meta-analysis??

• Does this UR not require approval of IRBs?

Reviewer #2: In general:

It is just very long and suggest to be shortened.

Introduction:

- Assessing the methodological quality of systematic review included in an umbrella review is part of the methodology and cannot be an objective. Please revise your second objective.

Method:

- The umbrella review required two authors to conduct each step of the process based on the PRISMA checklist. The data extraction process as you mentioned was conducted by one author only, while it should be conducted either in parallel or as confirmatory.

Results:

- The second sentence in line 186 (Total sample size…) It is not clear what you want to say, please rephrase.

Discussion:

- In line 353 NOF: Please do not start with writing an abbreviation without refereeing to the complete word, then you can repeat it freely.

- Possible mechanisms: This section is very interesting. I suggest to reduce the long discussion, removing all the repetition from the result and add it to give it a better value.

- Strength and limitation section is very long and no added value. Consider revision.

- You keep repeating that you used the high-quality reviews, while all of your included studies are either low or very low quality, Consider revision.

6. PLOS authors have the option to publish the peer review history of their article (what does this mean?). If published, this will include your full peer review and any attached files.

Reviewer #1: **Yes: **Beyene WA,

Reviewer #2: No

---

## [Author Response · Author response to Decision Letter 0]

4 Oct 2021

Thank you for the opportunity to resubmit a revised version of our manuscript now entitled ‘Dietary risk factors for hip fracture in adults: An umbrella review of meta-analyses of prospective cohort studies’. We thank all reviewers for their constructive comments and suggestions, and present point-by-point responses below. References to changes made refer to the revised manuscript with track changes. All responses below are also presented in the 'response to reviewers' letter. 

Response to reviewer #1

1. General

Reviewer comments: Does this UR not require approval of IRBs?

Response: To our knowledge, IRB approval is not required for this umbrella review since all analyses are based on published aggregated data, without collecting, storing, or presenting any individual participant data. 

2. Title

Reviewer comments

a) In articulating a title of this particular Umbrella Review, the population of interest is not evident. 

b) Is it a review of Meta-analysis (which is only 2 out of 34-exposures)?

c) Suggestion: Dietary risk factors for hip fracture in adults: An Umbrella Review of Prospective cohort studies

Response: Thank you for the suggestion. We have changed the title to ‘Dietary risk factors for hip fracture in adults: An umbrella review of meta-analyses of prospective cohort studies’ (lines 1 – 2 of the revised manuscript). 

3. Objective

Reviewer comments

a) ‘To summarize the totality of evidence regarding dietary factors and hip fracture incidence in adults, and to evaluate the methodological quality of systematic reviews and meta-analyses.’ Is it appropriate to use the Term `Incidence`?

b) `Methodological quality of systematic reviews` -This is the analysis procedure to be performed during the process and not the objective to be set. 

c) Suggestion: To summarize the `totality of evidence regarding dietary factors for hip fracture` in adults. 

Response: Thank you. We have rephrased the end of our introduction as aims rather than objectives (page 5, lines 82 – 84), and updated sections where our aims were repeated accordingly (abstract: page 2, lines 23 – 24; discussion: page 19, lines 298 – 299). We have removed evaluating methodological quality of reviews from our aims, but have kept evaluating the strength of evidence as an important aim of the review. 

4. Method

Reviewer comments

a) ‘The quality of evidence for all associations except combined fruits and vegetables intake and hip fracture incidence was downgraded from low to very low due to a serious or very serious risk of bias amongst primary studies`. The purpose of UR quality assessment is to assess methodological quality, risk of bias, and reporting quality of reviews, and does not involve retrieving the quality of primary studies??

b) Any modifications to the JBI umbrella review method?

Response

a) Quality assessment in our umbrella review was split into 1) methodological/ reporting quality of reviews (using AMSTAR-2); and 2) quality or strength of the evidence as a whole (using GRADE). To rate the quality of evidence for a given diet-hip fracture association, we used data presented in meta-analyses to account for factors such as risk of bias in primary studies and consistency amongst study effect sizes (overlap of confidence intervals) without re-analysing primary studies. These elements are not covered by AMSTAR-2, and it is important to consider bias at both the review and primary study levels when interpreting associations without unnecessarily revisiting primary studies. 

b) We did not aim to conduct a JBI review, but instead followed similar guidance from Fusar and Poli (2018) for conducting umbrella reviews [cited on line 89 of the revised manuscript, page 5], with presentation and reporting following PRISMA guidance. The step-by-step process was similar to the JBI method with some small differences, such as:

o Our protocol was registered on PROSPERO rather than JBI’s registration database. 

o We used AMSTAR-2 to assess methodological quality of reviews, whereas JBI has its own critical appraisal tool for JBI reviews. 

o The presentation style of results is similar to JBI reporting requirements, but does not follow them strictly. For example, JBI umbrella reviews must use colour-code summary tables according to the direction of effects using a traffic light system. We left summary tables colourless to avoid misleading the reader that a dietary factor might increase or decrease hip fracture risk when the evidence certainty was too low to tell. 

5. Results

Reviewer comments

a) `The quality of evidence (GRADE) was low for an inverse association between hip fracture incidence and intake of fruits and vegetables combined (adjusted relative risk for higher vs lower intakes: 0.92 [95% confidence interval: 0.87 to 0.98]), and very low for the remaining thirty-three exposures. Adjusted summary effect estimates and the quality of evidence for each exposure regarding risk of hip fracture are summarized.

o Question: Wasn`t it necessary to show cumulative risk factor (effect size)? By forest plot, Summary ROC (SROC) curves…or other ways of addressing heterogeneity of the Meta-analysis. Because, if the differences are not the result of chance, then we need to be cautious in interpreting the results of the meta-analysis.

b) Nineteen meta-analyses had an I2 value above 50%, and heterogeneity was significant for the 14 exposures. If studies are heterogeneous, is it certain or appropriate to synthesize the respective studies into meta-analysis??

Response

a) In the summary tables (Tables 1 – 4), we report effect estimates for each exposure (potential risk factors) using the aggregate effect estimates reported in meta-analyses (summary hazard ratios, relative risks, or odds ratios). We did not perform a meta-analysis of meta-analyses, nor did we identify any eligible reviews of cumulative risk factors acting synergistically to influence risk of hip fracture. The quoted lines (38 – 41 in the revised manuscript, page 2) have been minorly revised to include the term ‘adjusted summary relative risk for higher vs lower intakes…’

b) We agree that many of the meta-analyses included in our review of reviews combined effect estimates from primary studies when this was less appropriate – this is reflected in our quality assessment of reviews (Table S5, AMSTAR-2, item 11). For example, several included meta-analyses synthesised effect estimates from cohort studies that defined ‘high’ and ‘low’ intakes of a given dietary factor differently, without standardising exposure categories. In these cases, a score of 0 was awarded for item 11 to highlight a potential critical flaw in the review method. 

Response to reviewer #2

1. General comments

Reviewer comments: In general, it is juts very long and suggest to be shortened. 

Response: The revised manuscript has been shortened where possible whilst at the same time incorporating both reviewers’ helpful additions. The overall word count has been reduced from 6300 words to 5885 words, with the discussion word count dropping from 2615 words to 2215 words. We emphasise that PLOS ONE manuscripts can be any length and that umbrella reviews are naturally long due to the volume of material covered, but we have endeavoured to be concise throughout. 

2. Objectives

Reviewer comments: Assessing the methodological quality of systematic review included in an umbrella review is part of the methodology and cannot be an objective. Please revise your second objective. 

Response: Thank you. Please refer to the response to reviewer #1 under section 3 (objectives) where we address this point. 

3. Methods

Reviewer comments: The umbrella review required two authors to conduct each step of the process based on the PRISMA checklist. The data extraction process as you mentioned was conducted by one author only, while it should be conducted either in parallel or as confirmatory. 

Response: Thank you. Two reviewers (JW and CR) independently conducted title and abstract screening, full-text screening, extraction of outcome data (agreement level of 83%), assessments of the methodological quality of reviews (AMSTAR-2), and certainty of evidence assessments of associations (GRADE) in parallel. Disagreements at all stages were resolved by consensus. Study characteristics were extracted by one reviewer (JW). We then followed the PRISMA checklist by reporting our methods for each section transparently. The revised manuscript now clarifies that extraction of outcome data was conducted in parallel by two reviewers (page 7, lines 147 – 149). 

4. Results

Reviewer comments: The second sentence in line 186 (Total sample size…) It is not clear what you want to say, please rephrase. 

Response: The sentence reading ‘Total sample size…’ (page 10, lines 234 – 235 of the revised manuscript) has been rephrased to ‘The total sample size in each meta-analysis ranged from…’ for clarity. 

5. Discussion

Reviewer comments

a) In line 353 NOF: Please do not start with writing an abbreviation without refereeing to the complete word, then you can repeat it freely.

b) Possible mechanisms: This section is very interesting. I suggest to reduce the long discussion, removing all the repetition from the result and add it to give it a better value.

c) Strength and limitation section is very long and no added value. Consider revision.

d) You keep repeating that you used the high-quality reviews, while all of your included studies are either low or very low quality, Consider revision.

Response

a) ‘NOF’ has been replaced with ‘National Osteoporosis Foundation’ on line 357 (page 21 of the revised manuscript). 

b) Thank you. We have reduced unnecessary repetition of results and refocused on the potential mechanisms (pages 22 – 24, lines 393 – 446 of the revised manuscript). 

c) Thank you. Whilst we have retained discussion of the strengths and limitations of our work, we have reduced its length by 260 words (pages 24 – 26, lines 447 – 528 of the revised manuscript). 

d) Thank you. Repetition of using the higher quality meta-analyses has been removed, and a brief explanation for differences in the results of meta-analyses regarding dairy products has been added (page 21, lines 338 – 339).

---

## [Decision Letter · Decision Letter 1]

14 Oct 2021

Dietary risk factors for hip fracture in adults: An umbrella review of meta-analyses of prospective cohort studies

PONE-D-21-20733R1

Dear Dr. James Webster,

We’re pleased to inform you that your manuscript has been judged scientifically suitable for publication and will be formally accepted for publication once it meets all outstanding technical requirements.

Kind regards,

Yuan-Pin Hsu

Academic Editor

PLOS ONE

Additional Editor Comments (optional):

Reviewers' comments:

Reviewer's Responses to Questions

**Comments to the Author**

1. If the authors have adequately addressed your comments raised in a previous round of review and you feel that this manuscript is now acceptable for publication, you may indicate that here to bypass the “Comments to the Author” section, enter your conflict of interest statement in the “Confidential to Editor” section, and submit your "Accept" recommendation.

Reviewer #1: All comments have been addressed

Reviewer #2: All comments have been addressed

2. Is the manuscript technically sound, and do the data support the conclusions?

Reviewer #1: Yes

Reviewer #2: Yes

3. Has the statistical analysis been performed appropriately and rigorously? 

Reviewer #1: Yes

Reviewer #2: Yes

4. Have the authors made all data underlying the findings in their manuscript fully available?

Reviewer #1: Yes

Reviewer #2: Yes

5. Is the manuscript presented in an intelligible fashion and written in standard English?

Reviewer #1: Yes

Reviewer #2: Yes

6. Review Comments to the Author

Reviewer #1: (No Response)

Reviewer #2: (No Response)

7. PLOS authors have the option to publish the peer review history of their article (what does this mean?). If published, this will include your full peer review and any attached files.

Reviewer #1: **Yes: **Dr.Beyene WA

Reviewer #2: No

---

## [Editor Report · Acceptance letter]

19 Oct 2021

PONE-D-21-20733R1 

Dietary risk factors for hip fracture in adults: An umbrella review of meta-analyses of prospective cohort studies 

Dear Dr. Webster:

I'm pleased to inform you that your manuscript has been deemed suitable for publication in PLOS ONE. Congratulations! Your manuscript is now with our production department. 

Kind regards, 

on behalf of

Dr. Yuan-Pin Hsu 

Academic Editor

PLOS ONE